# Taurine Administration Counteracts Aging-Associated Impingement of Skeletal Muscle Regeneration by Reducing Inflammation and Oxidative Stress

**DOI:** 10.3390/antiox11051016

**Published:** 2022-05-21

**Authors:** Alessandra Barbiera, Silvia Sorrentino, Damon Fard, Elisa Lepore, Gigliola Sica, Gabriella Dobrowolny, Luca Tamagnone, Bianca Maria Scicchitano

**Affiliations:** 1Sezione di Istologia ed Embriologia, Dipartimento di Scienze della Vita e Sanità Pubblica, Università Cattolica del Sacro Cuore, L.go Francesco Vito 1, 00168 Roma, Italy; alessandra.barbiera@unicatt.it (A.B.); silvia.sorrentino@unicatt.it (S.S.); damon.fard@unicatt.it (D.F.); gigliola.sica@unicatt.it (G.S.); luca.tamagnone@unicatt.it (L.T.); 2Fondazione Policlinico Universitario A. Gemelli IRCCS, L.go Francesco Vito 1, 00168 Roma, Italy; 3DAHFMO-Unità di Istologia ed Embriologia Medica, Sapienza Università di Roma, 00161 Roma, Italy; elisa.lepore@uniroma1.it (E.L.); gabriella.dobrowolny@uniroma1.it (G.D.)

**Keywords:** nutrition, PGC1α, ROS, sarcopenia

## Abstract

Sarcopenia, which occurs during aging, is characterized by the gradual loss of skeletal muscle mass and function, resulting in a functional decline in physical abilities. Several factors contribute to the onset of sarcopenia, including reduced regenerative capacity, chronic low-grade inflammation, mitochondrial dysfunction, and increased oxidative stress, leading to the activation of catabolic pathways. Physical activity and adequate protein intake are considered effective strategies able to reduce the incidence and severity of sarcopenia by exerting beneficial effects in improving the muscular anabolic response during aging. Taurine is a non-essential amino acid that is highly expressed in mammalian tissues and, particularly, in skeletal muscle where it is involved in the regulation of biological processes and where it acts as an antioxidant and anti-inflammatory factor. Here, we evaluated whether taurine administration in old mice counteracts the physiopathological effects of aging in skeletal muscle. We showed that, in injured muscle, taurine enhances the regenerative process by downregulating the inflammatory response and preserving muscle fiber integrity. Moreover, taurine attenuates ROS production in aged muscles by maintaining a proper cellular redox balance, acting as an antioxidant molecule. Although further studies are needed to better elucidate the molecular mechanisms responsible for the beneficial effect of taurine on skeletal muscle homeostasis, these data demonstrate that taurine administration ameliorates the microenvironment allowing an efficient regenerative process and attenuation of the catabolic pathways related to the onset of sarcopenia.

## 1. Introduction

Aging is characterized by the gradual impairment of the principal physiological and biochemical functions of organs and tissues, and it is often associated with a progressive loss of skeletal muscle mass and strength, a condition known as sarcopenia [1]. The mechanisms responsible for sarcopenia are not completely understood; nevertheless, it is likely the result of multifactorial events including a compromised regenerative capability [2,3], chronic inflammation [4,5], increased levels of oxidative stress [5,6], and mitochondrial dysfunctions [7].

Muscle regeneration is a coordinated process in which satellite cells, the stem cell compartment of skeletal muscle, are activated to maintain and preserve tissue structure and function upon damage [8].

The first phase of the regenerative process is characterized by myofiber necrosis due to an influx of extracellular calcium leading to proteolysis of the myofibrils [9,10]. This event results in the activation of a specific inflammatory response that leads to sequential invasion of muscle by different inflammatory cell populations [11]. The inflammatory response is followed by satellite cell activation and by the formation of regenerating fibers, which are morphologically distinguishable by the characteristic centralized nuclei [12,13]. However, an efficient regenerative program could be severely affected in the case of aging or pathological conditions, and the formation of extended fibrotic tissue may contribute to functional impairment [14,15]. Moreover, changes in inflammatory cytokines, growth factors, and metabolic signals in the aged skeletal muscle environment may affect satellite cell proliferation and/or activation upon myofiber injury [16]. It is known, indeed, that aging is associated with a low-grade inflammatory state, a condition known as “inflammaging”, characterized by slightly increased plasma levels of pro-inflammatory mediators, such as tumor necrosis factor α (TNFα) and interleukin 6 (IL-6), and the consequent activation of the NF-κB pathway [13]. Interestingly, NF-κB protein concentrations were found to be four-fold higher in elderly human muscles compared to those of young people; this increased concentration is accompanied by anabolic signaling deficits resulting in the wasting of aged muscle [17].

Increased levels of inflammation are closely related to oxidative damage, and both are involved in the age-related reduction in muscle mass and strength. Oxidative stress is characterized by high levels of reactive oxygen species (ROS) and/or reactive nitrogen species (RNS). It can be caused by decreased antioxidant capacity due to impaired antioxidant enzymes activity and/or by increased ROS production [18]. In addition, elevated levels of ROS and RNS can also result as a consequence of mitochondrial dysfunction caused by age-related mitochondrial DNA mutations, deletions, and damage [19,20,21]. ROS appear to function as second messengers for TNF-α in skeletal muscle, activating NF-κB either directly or indirectly [14].

In skeletal muscle, the transcriptional coactivator peroxisome proliferator-activated receptor-gamma coactivator-1α (PGC-1α) is one the most important molecules involved in the stimulation of mitochondrial biogenesis, the regulation of cellular oxidant–antioxidant homeostasis, the suppression of chronic inflammation, and muscle catabolism [22]. PGC-1α interacts with nuclear receptors and transcription factors to activate transcription of their target genes, and its activity is responsive to multiple stimuli including calcium ions, ROS, insulin, thyroid and estrogen hormones, hypoxia, ATP demand, and cytokines [23]. In particular, PGC-1α regulation of mitochondrial biogenesis involves its interaction with several nuclear transcription factors, including PPAR family members, nuclear respiratory factor (NRF)-1 and NRF-2, myocyte enhancer factor-2 (MEF2), and forkhead box protein O (FOXO) 1 [24,25]. The PGC-1α activation of NRF-1, 2 promotes the expression of numerous nuclear-encoded mitochondrial proteins, which directly stimulates mitochondrial DNA replication and transcription [23,26,27]. Moreover, PGC-1α, in cooperation with the MEF2C transcription factor, may also influence myofiber phenotypic profiles favoring the shift from fast MHC toward the more resistant slow MHC during aging [28,29].

In the last decade, several strategies such as physical activity and nutrition have been proposed to potentially attenuate skeletal muscle deterioration during aging. Indeed, physical exercise has been reported to attenuate sarcopenia and prevent body fat accumulation and inflammation [30,31,32]. In addition, dietary interventions targeting protein or antioxidant intake may have a positive effect on increasing muscle mass and strength [33]. It is known that the loss of muscle mass and function that occurs in the elderly involves a decreased food intake, which results in the attenuation of muscle protein synthesis as compared to younger people [34]. Consequently, nutrition, in particular amino acid supplementation, may represent an important approach to improving the anabolic response of the muscle during aging [35,36,37,38,39].

Taurine is a cysteine-derived semi-essential amino acid highly expressed in mammalian tissues. In skeletal muscle, where its levels decrease during aging, it plays an important role as an antioxidant and anti-inflammatory molecule [40,41]. Since taurine-depleted skeletal muscle exhibits several abnormalities in its morphology and function, resembling those that occur during aging [42], taurine supplementation may represent a promising strategy to counteract the negative effects of aging in skeletal muscle.

Here, we demonstrate that intraperitoneal taurine administration counteracts aging-associated impingement of skeletal muscle regeneration by reducing inflammation. In addition, our results support the role of taurine as an anti-oxidant molecule able to ameliorate the muscle microenvironment, counteracting degenerative processes and favoring tissue homeostasis during aging.

## 2. Materials and Methods

### 2.1. Animals and Treatments

Young (8–10 weeks) and aged (18–20 months) male C57BL6J mice were housed in a facility with a light/dark cycle of 12 h at constant temperature and humidity. The mice were allowed to feed and drink ad libitum. The mice were treated according to the guidelines of the Ethics Committee of the Catholic University of the Sacred Heart—Rome (Authorization No. 150/2017-PR Italian Ministry of Health) in compliance with national regulations on the protection of animals used for scientific purposes (Italian decree no. 26 dated 4 March 2014, acknowledging European Directive 2010/63/EU). Taurine was prepared in a saline solution and administered via intraperitoneal injections at doses of 100 mg/kg/day for five consecutive weeks [43,44,45] (Figure 1). This dose was chosen based on published data showing antioxidant effects in in vivo mouse models [46,47]. The control mice received saline only. Before the induction of TA damage with cardiotoxin (CTX) injections, the animals were anesthetized through an intraperitoneal injection of a mix of ketamine 70 mg/kg and medetomidine 1 mg/kg, diluted in a physiological solution. An injury on the left-side tibialis anterior (TA) muscles of the control and taurine-treated mice was performed along the entire length of the muscle with four CTX injections (5 μL of 10 μM CTX per site) [48,49]. The right-side TA was used as a control counterpart. The animals were sacrificed through cervical dislocation after anesthesia as described above. For the histological analyses, the TA muscles were embedded in the OCT compound (Miles, Elkhart, IN, USA) and frozen immediately in isopentane at −80 °C.

### 2.2. Histological and Histochemical Analysis

The TA muscles of the old mice were sectioned at a thickness of 10 µm by a Leica cryostat. For the histological analysis, sections were stained with hematoxylin and eosin (H&E) using standard methods [50]. Esterase staining was performed using a nonspecific esterase stain kit (Bio-Optica, Milan, Italy) following the manufacturer’s instructions.

### 2.3. Morphometric Analysis

Hematoxylin and eosin and esterase staining were performed on sections of the TA samples. For the morphometric evaluation of fiber size, the analysis was performed on 4 randomly chosen fields of high-magnification images of whole muscle cross-sections for each condition. The number of examined animals was 3–4 for each treatment. The photomicrographs of the fibers were analyzed using ImageJ, Scion Image software (version beta 4.0.2; Scion Corporation, Frederick, MD, USA, http://rsb.info.nih.gov/ij, accessed on 2 May 2022) to evaluate the fibers’ cross-sectional area. The regenerating fibers were highlighted by the presence of central nuclei.

### 2.4. Immunofluorescence Analysis

Frozen sections were fixed in 4% paraformaldehyde for 10 min at room temperature, washed with PBS, permeabilized by a solution containing 1% bovine serum albumin (BSA) (Sigma-Aldrich, St. Louis, MO, USA, #A9418), 0.2% Triton-X in phosphate-buffered saline (PBS) for 30 min at room temperature, and blocked with 10% donkey serum (Sigma-Aldrich, St. Louis, MO, USA, #D9663) for 1 h at room temperature. The sections were incubated overnight at 4 °C with primary antibodies at the appropriate dilution. The following antibodies were used: slow MHC (Sigma-Aldrich, St. Louis, MO, USA, #M8421, 1:500) and 4-HNE (Alpha Diagnostics International, San Antonio, TX, #HNE11-S, 1:500). After being washed three times in PBS, the sections were then incubated for 60 min at room temperature with specific secondary antibodies. In particular, the following were used: AlexaFluor594-conjugated anti-mouse 1:1000 (Molecular Probes, Eugene, OR, USA, #A21203) and AlexaFluor488-conjugated anti-rabbit 1:1000 (Molecular Probes, Eugene, OR, USA, #A21206) in PBS containing 1.5% donkey serum. The sections were mounted with ProLong™ Gold Antifade Mountant with DAPI (Thermo Fisher Scientific, Waltham, MA, USA, #P36935) and examined with a Leica SP5 Laser confocal. Quantification of the changes in the 4-HNE signal in the experimental groups was performed by densitometric analyses. After background subtraction, the 4-HNE fiber-associated signals were quantified by manually outlining individual fibers and measuring fiber-associated fluorescence intensity with the ImageJ software. The F/A ratio defines the mean fluorescence of individual fibers (F) normalized to total fiber cross-sectional area (A). Quantification was performed on 50 fibers per group (*n* = 3 mice per group).

### 2.5. Protein Extraction and Western Blot Analysis

The TA muscles obtained from the mice were dissected, minced, and homogenized with lysis buffer (Cell Signaling Technology, Danvers, MA, USA, #9803) containing phenylmethylsulfonyl fluoride (PMSF) (Cell Signaling Technology, Danvers, MA, USA, #8553) and a complete protease inhibitor cocktail (Cell Signaling Technology, Danvers, MA, USA, #5872). The Bradford Protein Assay (Bio-Rad Laboratories Inc., Hercules, CA, USA) and Varioskan™ LUX controlled by Thermo Scientific™ SkanIt™, for Microplate Readers (Thermo Fisher Scientific, Waltham, MA, USA, Software version 4.1) were used to determine an equal amount of proteins according to the manufacturer’s instructions. The proteins were separated by SDS/PAGE (Mini-PROTEAN^®^ TGX™ Precast Protein Gels or Mini-PROTEAN TGX stain-free precast PAGE gels; Bio-Rad Laboratories Inc., Hercules, CA, USA) and transferred to a nitrocellulose membrane (Trans-Blot^®^ Turbo™ Mini Nitrocellulose Transfer Packs #1704158; Bio-Rad Laboratories Inc., Hercules, CA, USA). Nonspecific binding was blocked in Tris-buffered saline (TBS) (Bio-Rad Laboratories Inc., Hercules, CA, USA) supplemented with 0.1% Tween-20 and containing 5% nonfat dry milk (Bio-Rad Laboratories Inc., Hercules, CA, USA #1706404) for 1 h at room temperature. The primary antibodies used were: mouse monoclonal anti-SOD-1 (1:500, Santa Cruz Biotechnology Dallas, Texas 75220 U.S.A., sc-17767); rabbit monoclonal anti-phospho-mTOR (1:1000, #2971, Cell Signaling Technology, Danvers, MA, USA); rabbit monoclonal anti-mTOR (1:1000, #2972, Cell Signaling Technology, Danvers, MA, USA); mouse monoclonal anti-slow-MHC (1:500, Sigma-Aldrich, #M8421); mouse monoclonal anti-myosin (Skeletal, Fast) (1:500, Sigma-Aldrich, St. Louis, MO, USA, #M4276); rabbit monoclonal anti-phospho-NF-κB p65 (Ser468) (1:1000, #3039, Cell Signaling Technology, Danvers, MA, USA); rabbit monoclonal anti-NF-κB p65 (1:250, #3034, Cell Signaling Technology, Danvers, MA, USA); mouse monoclonal anti-G6PD (1:300, Santa Cruz Biotechnology Dallas, Texas 75220 U.S.A., sc-373887), and mouse monoclonal anti-GP91[phox] (1:300, BD Transduction Laboratories, United States http://www.bdbiosciences.com (accessed on 2 May 2022) #611414). The blots were then incubated with the following secondary antibodies from Bio-Rad Laboratories: Goat anti-Rabbit IgG (1:3000, HRP Conjugate, Bio-Rad Laboratories Inc., Hercules, CA, USA, #1706515) and Goat anti-Mouse IgG (1:3000, HRP Conjugate, Bio-Rad Laboratories Inc., Hercules, CA, USA, #1706516) for 1 h at room temperature. Signals were captured by ChemiDoc™ Imaging System (Bio-Rad Laboratories, Hercules, CA, USA) using an enhanced chemiluminescence system (SuperSignal Chemoluminescent Substrate, Thermo Fisher Scientific Inc. Waltham, MA, USA). Densitometric analyses were performed using Image Lab™ Touch, Software version 5.2.1 (Bio-Rad Laboratories Inc., Hercules, CA, USA ).

### 2.6. Real-Time PCR Analysis

The TA muscles obtained from the mice were dissected, and total RNA extraction was performed with a tissue lyser (QIAGEN) in TriReagent (Invitrogen, Carlsbad, CA, USA) according to the manufacturer’s protocol. Double-stranded cDNA was synthesized with the QuantiTect Reverse Transcription kit (QIAGEN S.r.l., Milan, Italy,). Messenger-RNA analyses were performed on an ABI PRISM 7500 SDS (Applied Biosystems, Waltham, MA, USA) using specific TaqMan assays (Applied Biosystems, Waltham, MA, USA). Specifically, the following assays were used: GPX1: mm_00656767_g1; MEF2C: mm_00600423_m1; PGC1-α: mm_01280835_m1; SOD1: mm_01344233_m1; and CAT: mm_00437992_m1. Relative quantification was performed using the endogenous control Hprt1 (Applied Biosystems, Waltham, MA, USA). Real-time PCR was performed using RNA preparations from three to five different animals for each group as specified in the Figures. The relative expression was calculated using the 2−∆∆Ct method.

### 2.7. Statistical Analysis

Multiple statistical comparisons between groups were performed by one-way ANOVA. Where indicated in the legends, the Mann–Whitney rank-sum test or unpaired Student’s *t*-test were used. Each bar represents the mean ± SEM (standard error of the mean).

## 3. Results

### 3.1. Taurine Administration Counteracts Aging-Associated Impingement of Skeletal Muscle Regeneration

To investigate the effect of taurine administration on skeletal muscle regeneration, we induced damage by means of CTX injections in the TA muscles of the control (vehicle) and taurine-treated old mice. The morphological and morphometric analyses revealed that, in the absence of injury (Figure 1A,B: panels a, b and Figure 1C), the muscle fibers of the taurine-treated mice displayed a slightly increased cross-sectional area compared to the controls. Since protein homeostasis in skeletal muscle rests on an equilibrium between protein synthesis and protein degradation, we then analyzed the levels of both phospho-mTOR, as the main regulator of protein synthesis, and Atrogin-1, as one of the major regulators involved in protein catabolism through the ubiquitin–proteasome system [40]. Our results showed that the levels of phospho-mTOR were significantly increased in the muscle extracts of the taurine-treated mice, while no significant modulation of Atrogin-1 (FBXO32) was revealed in this condition (Figure 1D–F). These data demonstrate the involvement of the mTOR-dependent pathway in the effect of taurine on the observed increase in skeletal muscle fiber CSA. In addition, after 1 week of muscle damage, the cross-section of the fibers in the vehicle-treated mice (CTX) revealed degeneration with concomitant acute inflammation and necrosis, as well as the presence of small regenerating fibers, identified by central nuclei (Figure 1G,H: panel c). On the other hand, larger regenerating myofibers and fewer necrotic fibers (Figure 1G,H: panel d) appeared in the injured muscles of the taurine-treated old mice, along with fewer infiltrates. The analysis of these results, shown in the diagrams in Figure 1I, revealed that the taurine administration affected the fiber size distribution by favoring the accumulation of larger regenerating fibers compared to the control injured muscle. In conclusion, these results suggest that taurine is able to stimulate the regenerative process by exerting a protective role in the maintenance of the skeletal muscle fibers’ integrity and by favoring the acceleration of the formation of new myofibers.

### 3.2. Taurine Supplementation Modulates the Inflammatory Response in Aged Muscle

Aging is accompanied by a chronic low-grade systemic inflammatory state [51] that may be responsible for the impaired regenerative capacity of skeletal muscle [52]. To verify whether the enhanced regenerative response observed in the injured muscles of the taurine-treated mice was associated with a modulation of the inflammatory process, we examined the presence of macrophages by esterase staining. Figure 2A,B show that, as a result of the CTX injections, all the muscle sections displayed an increased number of mononucleated inflammatory cells compared to the uninjured counterparts; however, the high number of esterase-positive macrophages present in old injured muscle (Figure 2A: panel c and Figure 2B) was significantly attenuated in the presence of taurine supplementation (Figure 2A: panel d and Figure 2B). The effect of taurine on decreasing the extent of inflammation during the regenerative process was also evaluated by analyzing the levels of the phosphorylated isoform of the transcription factor NF-kB since it is known that its activation in skeletal muscle leads to the degradation of specific muscle proteins, induces inflammation and fibrosis, and blocks the regeneration of myofibers after injury [53,54,55]. As shown in Figure 2C,D, phospho-NF-kB was detectable at very low levels in both the control and taurine-treated uninjured muscles, while the high levels of phospho-NF-kB detected in CTX injured muscles were dramatically decreased in the muscles of taurine-treated old mice. The total NF-kB levels and the ratio between phospho-NF-kB and NF-kB were analyzed and proved to have increased, albeit not significantly, with CTX-induced damage, while taurine prevented this effect (Figure 2C,E,F). These results demonstrate that taurine attenuates the inflammatory status in injured muscle by decreasing the levels of both total NF-kB and phospho-NF-kB.

### 3.3. Taurine Modulates PGC1-α and MEF2C Expression in the TA Muscles of Aged Mice

To better investigate the molecular mechanisms involved in the positive effect of taurine on skeletal muscle homeostasis, we evaluated the role of the transcriptional co-activator PGC-1α. PGC-1α is an important factor promoting an anti-inflammatory environment in muscle. In addition, it has been reported that PGC-1α may improve not only muscle function but also myofiber morphology and integrity, implying a potential role for PGC-1α in fiber repair and regeneration. In cooperation with the MEF2C transcription factor, PGC-1α has been shown to regulate skeletal muscle fiber-type determination, promoting the switch from glycolytic fibers to the more resistant oxidative ones [56,57].

Thus, using an RT-PCR analysis, we evaluated the mRNA expression levels of PGC-1α and MEF2C in the TA extracts of young, old, and old taurine-treated mice to determine whether taurine administration induced transcriptional changes of the abovementioned factors as compared to what was observed in its absence. Young mice in a healthy condition were used to assess the expression levels of these factors. As shown in Figure 3A,B, in the absence of taurine, no changes in PGC-1α levels and only a slight decrease in MEF2C levels were observed in the extracts of old mice compared to the young ones, while the mRNA levels of both molecules were significantly upregulated in the old mice subjected to intraperitoneal taurine injections. It has been demonstrated that the type I slow-twitch oxidative fibers (expressing the slow isoform of the myosin heavy chain, slow MHC) are more resistant to damage and a variety of atrophic conditions than type IIb fast-twitch glycolytic fibers [29]. In several muscle pathologies, including sarcopenia, the fastest muscle phenotype is more severely compromised when compared with slow-twitch muscles, and the greater sensitivity of the type IIb fibers may be due to their lower content of PGC-1α compared to that of the oxidative fibers [58,59]. Here, we showed by Western blot analysis that the TA muscles of the old mice expressed very low levels of the slow-MHC isoform compared to the young-derived muscle extracts; however, slow-MHC expression increased in the muscle extracts of the taurine-treated mice (Figure 3C,D). In addition, the Western blot analysis of the fast-MHC isoform revealed that, in the presence of taurine, its expression was significantly upregulated compared to what was observed in the TA extracts derived from old mice that did not receive taurine administration (Figure 3F). Consistently, the reduced levels of MHC (MF20) detected in the muscle extracts of old mice, as compared to those in the young mice, were increased with taurine administration. These results suggest that the positive effect of taurine on skeletal muscle homeostasis of aged mice may be mediated by the stimulation of the PGC1-α/MEF2C pathway, favoring a possible metabolic shift of the myofibers towards the oxidative phenotype and preserving the more susceptible glycolytic fibers.

### 3.4. Taurine Attenuates Oxidative Stress in TA Muscles of Aged Mice

Age-related sarcopenia is often associated with enhanced ROS production [5]. Taurine has been found at particularly high concentrations in tissues exposed to elevated levels of oxidants, suggesting a role in the attenuation of oxidative stress [40,60,61]. Thus, we evaluated whether the effect of taurine in skeletal muscle homeostasis of aged mice was correlated to the modulation of oxidative stress. To this purpose, we analyzed the levels of the Gp91phox protein, the catalytic subunit of the enzymatic complex nicotinamide adenine dinucleotide phosphate (NADPH) oxidase 2 (NOX2), responsible for the conversion of molecular oxygen to superoxide (O_2_^−^) [62,63]. A significant increase in the Gp91phox protein level was observed in the muscles of the old mice compared to that present in the young mice (Figure 4A,B), highlighting an age-related generation of superoxide in older muscles. However, in the old mice treated with taurine, the expression of the Gp91phox protein returned to levels comparable to those of the young group. Another molecule involved in the maintenance of cellular levels of NADPH with both pro- and antioxidant activity is glucose-6-phosphate dehydrogenase (G6PD), whose altered level has been described as a consequence of NO signaling dysregulation [64,65]. We observed a significant increase in the G6PD protein in TA muscles of the old mice compared to the young group, while the presence of high levels of taurine reduced G6PD to levels comparable to those of the young group (Figure 4A,C). These data suggest that taurine can counteract the deregulation of redox-related circuits, and consequently decreases NOX2-dependent ROS production.

To confirm the role of taurine as an anti-oxidant molecule, using a real-time PCR analysis, we further analyzed the expression levels of several antioxidant genes such as SOD1, CAT, and GPX1, known to undergo upregulation as a consequence of increased ROS production during aging [66]. As shown in Figure 4D–F, all the molecule expression levels were upregulated in the muscles of the old mice compared to those of the young group, but when the mice were treated with taurine, SOD1, CAT, and GPX1 expression was reduced, reaching a level comparable to what was found in the young mice-derived muscle extracts. Moreover, using Western blot analysis, we analyzed the levels of SOD in the different experimental groups (as indicated in the Figure), showing that the increased levels of SOD observed in the muscle extracts of the old mice were significantly reduced in the presence of taurine, confirming the results of the RT-PCR analysis (Figure 4G,H). These results demonstrate an important role played by taurine in the attenuation of the elevated level of oxidative stress that characterizes aged muscle. Consistently, taurine reduced ROS accumulation detected in the TA muscles of old mice (Appendix A) [67]. To investigate whether the abundance of ROS detected can induce oxidative modification of proteins, we performed an immunofluorescence analysis using 4-hydroxy-2-nonenal (4-HNE) adducts as markers of damage or alteration of muscle proteins due to oxidative stress [68]. Our results, shown in Figure 4I,J, demonstrated that 4-HNE expression was higher in the TA muscles of old mice, where slow MHC was also strongly downregulated; in the presence of taurine, 4-HNE expression was significantly decreased with a concomitant slow-MHC upregulation. These results are in agreement with our previous data (see Figure 3C,D) and strongly suggest taurine’s important role in the attenuation of ROS accumulation, preserving the slow-fiber phenotype during aging.

## 4. Discussion

In our previous studies, we demonstrated that taurine exerts a positive effect on myogenic differentiation and homeostasis in cell cultures [33]. Here, we investigated its effects in an in vivo experimental model. For this purpose, we used aged mice in which taurine was intraperitoneally injected every day for 5 weeks in order to assess the impact of taurine in the modulation of processes, such as regeneration, inflammation, and oxidative stress, which are known to be dysregulated during aging. We demonstrated that taurine accelerated the regenerative process of CTX-damaged TA muscles, preserving the architecture of skeletal muscle tissue. Indeed, 7 days after damage induction, in the presence of high levels of taurine, we observed a lower amount of inflammatory infiltration and fibrosis, and the regenerating fibers appeared larger compared to those of the vehicle-treated control muscles. This effect seems to be mediated by taurine-dependent stimulation of anabolic pathways, as demonstrated by the increased levels of phospho-mTOR, rather than an effect on the modulation of catabolic processes; indeed, although the activation of other catabolic pathways cannot be excluded, ubiquitin ligase atrogin-1 is not significantly modulated by taurine. In general, skeletal muscle regeneration is guaranteed by the presence of satellite cells, whose number and activity significantly decrease during aging [69]. It has been demonstrated that the alteration of the immune response with aging, known as immunosenescence, is one of the main causes related to the hampered regenerative capability of skeletal muscle [70]. Indeed, immunosenescence promotes the development of a chronic low-grade inflammatory state, which may alter satellite cell proliferation and/or activity, thereby contributing to the impairment of the repairing capacity [69]. Thus, we verified whether the positive effect of taurine on skeletal muscle regeneration was mediated by the modulation of the inflammatory state. Here, we showed that the high number of macrophages present in old injured muscle was significantly decreased in the presence of taurine. This effect appeared to be mediated by NF-kB signaling since we showed that its elevated levels in CTX-injured muscles were decreased in the taurine-treated aged mice. These data are in agreement with what we have previously demonstrated in an in vitro experimental model [33] and are consistent with the role of taurine as an anti-inflammatory molecule exerting its effect, at least in part, through the inhibition of NF-kB activation [71]. In particular, it has been demonstrated that taurine may protect tissue damage from inflammation because its amino group can neutralize hypochlorous acid generated by inflammatory cells, downregulating the production of cytokines, and, finally, decreasing the immune response [72,73]. The chronic low-grade inflammatory state characterizing aged muscles may have a significant impact on the stimulation of catabolic pathways and mitochondrial dysfunctions, all contributing to the onset of sarcopenia [74]. In this context, the transcriptional co-activator PGC-1α appears to play a crucial role against skeletal muscle deterioration during aging. Indeed, it has been reported that PGC-1α plays a protective role in the inflammatory response, reducing pro-inflammatory cytokine production and exerting a regulatory mechanism for the expression of endogenous antioxidant proteins; moreover, it may improve muscle function, myofiber morphology, and integrity, suggesting its potential role in fiber repair and regeneration. Additionally, in cooperation with the MEF2C transcription factor, PGC-1α has been shown to regulate skeletal muscle fiber-type differentiation, promoting the switch from glycolytic fibers to the more resistant oxidative ones [56,57]. Here, we showed that, in the absence of damage, no changes in PGC-1α levels and only a slight decrease in MEF2C levels were detected in TA muscle extracts of old mice compared to what was observed in young animals; however, their expression was significantly increased in the presence of taurine, reaching levels comparable to those found in the TA muscles of the young group. In addition, our results showed that taurine increases the levels of total MHC (MF20) and the slow-MHC and fast-MHC isoforms, suggesting its potential role in the metabolic shift of aged skeletal muscle fibers towards the oxidative, more resistant, phenotype [29]. These data reveal that the positive effect of taurine on skeletal muscle homeostasis of aged mice may be mediated by the stimulation of the PGC1-**α**/MEF2C pathway, favoring a possible metabolic shift of the myofibers towards the oxidative phenotype and preserving the more susceptible glycolytic fibers.

Taurine has been found at particularly high concentrations in tissues exposed to elevated levels of oxidants [40,75,76], and this prompted us to evaluate whether the observed positive effect of taurine on aged skeletal muscle homeostasis was related to the modulation of oxidative stress. A crucial mediator of ROS production in skeletal muscle tissue is the Gp91phox protein, which represents the catalytic subunit of the NOX2 complex and is also known to be overexpressed in dystrophic conditions [62,63,77,78,79]. Thus, we analyzed the Gp91phox protein in our experimental models, revealing that its level, while strongly upregulated in old mice compared to young ones, is significantly reduced in the presence of taurine. NOX2-dependent O_2_^−^ production is closely correlated with the availability of NADPH, although this substrate is also part of the antioxidant system contributing to the neutralization of ROS. In this context, one of the crucial enzymes involved in the maintenance of the cellular levels of NADPH is G6PD, which has pro- or antioxidant activity in skeletal muscle [65]. Here, we reported that the enhanced level of G6PD observed in old mice is significantly reduced in the presence of taurine, supporting the role of taurine as a potent modulator of NOX2-dependent ROS production in aged skeletal muscle. As a confirmation of this hypothesis, we showed that the accumulation of ROS in old muscle (see Appendix A) was strongly decreased by treatment with high doses of taurine. This effect was accompanied by the diminished formation of 4-HNE protein adducts, which are considered markers of lipid peroxidation and altered cellular redox homeostasis. We also showed that the endogenous antioxidant response in aged skeletal muscle is modulated in presence of taurine, as revealed by the analysis of important anti-oxidant effectors such as SOD1, GPX1, and CAT. Indeed, the high levels of these molecules found in the TA muscle extracts of old mice were reduced upon taurine administration.

## 5. Conclusions

Collectively, our results show that, in aged muscle, taurine administration counteracts aging impingement of skeletal muscle regeneration, attenuates low levels of chronic inflammation, and decreases high levels of oxidative stress. Although the molecular mechanisms underlying these effects have not been completely elucidated, our data demonstrate that taurine administration ameliorates the microenvironment that allows the maintenance of skeletal muscle homeostasis and counteracts the aging process.

## Data Availability

Data is contained within the article.

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
