# Peer review of "Taurine Administration Counteracts Aging-Associated Impingement of Skeletal Muscle Regeneration by Reducing Inflammation and Oxidative Stress"

_antioxidants, 2022, doi:10.3390/antiox11051016_

Round 1
Reviewer 1 Report
In general terms, this study provides data indicating that five weeks of taurine treatment have several positive effects on aging skeletal muscle. However, methodological issues, selection of proteins/mRNAs to assess, and quality of results decrease the impact of the study, as well as the possibility to draw solid conclusions. Accordingly, the authors actually conclude “it is possible to speculate that taurine ameliorates the systemic regulations allowing the maintenance of skeletal muscle homeostasis and counteracting the aging process” (lines 429-431). Thus, the authors seem to admit that the study presents preliminary results, which can form the basis for speculations, but further studies are required for solid conclusions.
Specific points.
- Lines 104-111: The description of anesthesia, injections, and handling of TA muscles has to be improved. What is meant by “100 mg/Kg taurine”? Method for cardiotoxin injection needs more details, including anesthesia or some kind of pain relief. Were both legs injected with CTX; if not, was the other leg used as control? Method of killing? TA muscles used for Western blotting and PCR were not embedded in O.C.T.
- Morphometric analysis (lines 120-125 and Figure 1: More details on how fiber sizes were measured are required. How were boundaries of individual muscle fibers defined? How were the large areas lacking muscle fibers in the CTX group accounted for? Should not the sum of fiber distributions in Fig 1D equal 100% - now it looks like less than 100% in the CTX group and markedly more than 100% in the TAU+CTX group.
- ROS determination: CM-H2DCFDA is designed to be used to measure ROS production in living cells. Acetate groups (DA) make it possible for the indicator to pass the cell membrane. The acetate groups are then cleaved by intracellular esterases and the indicator becomes active. Thus, it is questionable whether this indicator will measure ROS in the present frozen muscle sections. Have the authors performed any control experiments or are there strong literature support for it being a valid method in frozen sections? If so, more methodological details are required.
- RT PCR analysis: The primers used need to be presented.
- Statistical analysis: Student´s t-test is not appropriate for most analyses, i.e. different types of ANOVA should be used. Repeated measurements are performed in Fig. 1D. Four groups are studied in Fig. 2C; three group are studied in Figs. 3 & 4. The number of muscles and mice used for each analysis needs to be stated in all legends (or was only one TA from each mouse used).
- Figure 2: The standard control for the extent of phosphorylation of a protein is assessment of the total amount of this protein, thus in this case Western blotting with an antibody that detects both phosphorylated and unphosphorylated NK-kB. Can this analysis be performed to replace GAPDH as the loading control? If the major difference in MHC distribution reported if Fig. 3 is true, differences in the glycolytic enzyme GAPDH might be expected.
- Lines 249-250: This conclusive heading is not supported by the data presented, which at best show an association.
- Lines 269-283 and Fig. 3C & D: The marked difference in slow-MHC expression between TA muscles of old untreated and taurine-treated mice is remarkable. Is it likely that only five weeks of treatment will have this marked effect on fiber type composition, which generally is very robust and requires severe manipulations to occur? Any literature support for this finding? Or does the result reflect the fact that TA muscles contain very little slow-MHC and hence large relative changes would have little impact on the overall fiber type composition. The accompanying text only mentions type I and IIb fibers – what about type IIa and IIx fibers? A solid conclusion on TA fiber type composition between untreated and taurine-treated mice requires some measure also of type II fiber expression.
- Assessment of oxidative stress: The rationale for using the protein expression of Gp91phox and G6PD needs to be discussed in greater detail. The discussion on this point starts with identifying mitochondrial dysfunction as a major cause of increased ROS production in in aged muscle (lines 399-401). It then seems odd that proteins not related to mitochondria were measured to assess oxidative stress. Moreover, the mRNA levels of enzymes that are part of the cellular defense against oxidative stress were higher in untreated that in taurine-treated muscles. If this difference in mRNA levels translate into similar differences in protein levels, untreated muscles would have a higher antioxidant capacity than taurine-treated muscle, hence being opposite to the present suggestions. Western blotting to measure the protein expression of SOD1, GPX1 and CAT are required to resolve this fundamental issue.
Author Response
Please find the attached PDF file including our responses to your comments.

Reviewer 2 Report
Manuscript ID: antioxidants-1668028
I have read with a great interest the paper entitled “Taurine administration counteracts aging-associated impingement of skeletal muscle regeneration by reducing inflammation and oxidative stress” by Barbiera et al., and I have some comments about it.
In this paper, Barbiera and co-workers demonstrated that taurine injection have beneficial effects in old mice through improvement of skeletal muscle regeneration and the decrease of oxidative stress and inflammatory response. They have showed that the ROS production was attenuated, and that PGC1-α pathway could be involved in the phenotypic changes observed in aged mice receiving taurine.
The paper is of interest since it provides data about the beneficial effect of taurine toward sarcopenia. However, some data need to be provided to be fully enthusiasm about this paper.
Major comments
- Why tibialis was chosen? Did authors have some data about the impact of taurine on slow muscles such as soleus?
- TA is a fast muscle. Why slow MHC was preferentially analysed instead of fast isoforms?
- Arrows or other symbols should be added to help the reader to observe central nuclei fibers and necrotic fibers on histological images of Fig.1. The same remark is done for figure 2.
- Student-t test is not well adapted since there are more than two groups that are compared. A stronger statistical test have to be applied.
- The increase of CSA can result from increase of protein synthesis and/or protein degradation. Since taurine may impact the Akt/mTOR pathway, it could be interesting to determine whether mTOR pathway is modulated (phosphorylation status of p70S6K or 4EBP1) or degradation pathway (FoxO or atrogens expression such as MurF1 or MAFBx). This point should be at least discussed even if no additional data are provided.
- 2: fibers seem to be differentially stained. Is there a correspondence with the phenotype of the fibers? Did authors determine whether the typology of TA was changes? Are the different types of fibers impacted in the same manner by taurine treatment?
- Several aspecific signals for pNF-kB p65, and some signals are more intense than the signal indicated with arrow. Are authors sure about the signal corresponding to pNF-kB p65? The ratio phospho/total is usually used to consider the variation of phosphorylation. Did authors performed the blot against NF-kB to determine whether its expression could change? This important point need to be considered in the reviewed paper.
- GAPDH was used to verify the sample loading. Surprisingly, there is a lower molecular weight signal on blots revealed with GAPDH antibody. In addition, there is a strong signal on GAPDH blots at a lower molecular weight in CTX muscle. Is it known whether GAPDH could be affected consecutively to CTX treatment? In addition, the Stain-free acrylamide offers the great advantage to normalize the western blot quantification without using of housekeeping protein. In the same way, the blot on figure 2 and 3 were normalized with GAPDH, whereas on figure 4, the sample loading was verify with stain free signal. Why the method of normalization changed?
- Two bands are observed for the slow MHC bands. Which one corresponds to slow MHC? Moreover, the two bands are very close. Which one was quantified? Are authors sure that the quantified signal corresponds to slow MHC since the signals have a very similar intensity? Lastly, the merge between stain-free membrane and MHC lacks. Images need to be provided.
- The dramatic reduction (>95%) of slow MHC is a little bit surprising. Does it means that slow fibers are preferentially impacted? Perhaps the quantification of fast isoforms of MHC should be done.
- It is worth to note that there are strong differences between blot of series 1-2 and 3-4 for G6PD. Indeed, for the series 1-2, the signals are well defined with a high signal/noise ratio. However, for series 3-4, there is a strong background, rendering quantification difficult. The same antibody was used? The western blot protocol was the same?
Minor comments
- 12: the author(s) name are lacking.
- Some typographical errors that need to be corrected.
- Line 78: “…nutrition has been shown to be one of the main strategies potentially attenuating skeletal muscle deterioration during aging”. It is true, but exercise too. Please complete de sentence to take into account the beneficial effects of physical activity in aged population.
- Line 85: the term expression better fits with to protein. It should be replace
- 2A: is it same magnification for a and b panels?
- Mice series 2, slow MHC: the arrow indicating 223 kDa is located between the two bands.
- Scale lack on histological images.
Author Response
Please find the attached PDF file that includes our responses to your comments.

Reviewer 3 Report
This study sought to test whether taurine treatment is effective against age-associated defects in skeletal muscle regeneration by reducing inflammation and oxidative stress. Overall, the results were interesting and the following suggestions are proposed to improve the manuscript:
- In figure 1, panels C and D do not show all 4 treatment groups as panels A and B do. Restructuring the format could better highlight the true impact of taurine treatment.
- In Results section 3.1, the authors note that taurine protects muscle by inhibiting inflammation and necrosis while pointing to figure 1. However, figure 1 does not appear to show data related to inflammation and necrosis.
- Is it possible to quantify the results in figure 2A?
- In figure 3, assessing PGC-1alpha and MEF2C gives a fairly narrow view of the mechanism. Please elaborate on why RT-PCR is sufficient to understand these factors and their actions.
- Is it possible to quantify the results in figure 4i?
Author Response
Please find the attached PDF file related to our responses to your comments.

Round 2
Reviewer 1 Report
The authors have addressed all points of my original review and performed some additional experiments in response to my critique. Unfortunately, this raises additional concerns about the quality of Western blots and measurements performed on these:
- The increase in phospho-mTOR in the representative blot (Fig. 1D) seems much larger than the quantification of the extent of phosphorylation (Fig. 1E). Is this blot representative? Individual data points in addition to the mean data is recommended for all Western blot experiments and this would also clarify then number of measurements.
- The increases in total NF-kB in CTX muscles in the representative blot (Fig. 2C) seem much larger than the increases in mean data (Fig. 2E). Thus, the points raised above whether the blot is representative and addition of individual data points also apply here. The extent of phosphorylation is generally performed as the ratio of phosphorylated to total (as in Fig. 1E) and this is missing here. Any idea on what the dense bands at the bottom of the CTX muscles’ stain-free blots represent? They are not seen in the other stain-free blots, and it seems like they would have a large impact on the NF-kB quantifications.
- The MHC results are puzzling. Myosin and actin are the absolutely dominating proteins in skeletal muscle. It is then difficult to understand how both slow-MHC and fast-MHC can increase dramatically with 5 weeks of taurine treatment in old muscle. This would imply that taurine markedly increases the myofibrillar content in aged muscle, which would lead to a major increase in force generating capacity. Morevoer, some other protein components in the muscles must decrease to a similar extent. If true, this finding would have a very large impact on our understanding of protein changes in aged muscle and hence needs further assessment. The MHC bands in the stain-free gels in the figures could be used to assess whether the total myosin content was affected by taurine treatment; unfortunately, the MHC bands are generally difficult discern in the displayed stain-free blots. What about actin – did it also show some increase with taurine treatment?
In relation to the 3rd point of my original review, I recommend the authors to completely delete the CM-H2DCFDA data.
Author Response
Dear Reviewer, please find the attached file related to our response to your comments.

Reviewer 2 Report
Authors have modified the manuscript as required, and they also answered to several of my comments and questions. They have partly rewritten the introduction, completed the Materials and Methods section, and also quantified western blot considering normalization with the Stain free profile. In addition, they applied more stringent statistical test.
However, it remains some comments regardless to the revised manuscript.
- Authors have quantified the mRNA of one atrogen, suggesting that the degradative pathway through the ubiquitin-proteasome system was not involved. I think it would be of interest to include this data in the paper. In addition, since in skeletal muscle, protein homeostasis rests on an equilibrium between protein synthesis (mainly regulated by mTOR pathway) and protein degradation (involving the ubiquitin-proteasome system and the atrogens, the autophagy or the calpain), it could be interesting to discuss this point in the discussion section since one degradative pathway was preferentially considered, and to mention that it could not be excluded that another degradative pathway could be involved as well.
- Line 243: “This effect seems to be mediated by the stimulation of the phospho-mTOR, since its expression is significantly increased in the muscle extracts of taurine-treated mice”. In view of the data presented in figure 1, the expression, i.e. the protein level of mTOR seems to be unaffected by taurine treatment. However, the phosphorylation level is modified. This sentence needs to be modified by “[…] since its phosphorylation level is significantly increased […]”. A similar comment is done for line 289 with the phospho-NF-kB expression; it corresponds to the phosphorylation level of NF-kB, not its expression.
- Figure 1H is different to those in the submitted paper. It should be of interest to provide data about fibers distribution for control and taurine to compare with control condition.
- Lines 292-293: “Total NF-kB levels were analyzed and proved to increase consistently with the CTX-induced inflammation, while taurine prevented this change”. It is true for the phosphorylation of NF-kB; however, for total NF-kB (i.e. NF-kB expression), the results are not significantly different as observed on figure 2E. Moreover, according to western blot image on figure 2C, the expression of NF-kB seems to increase in CTX and CTX+Tau conditions, that is not correlated with the corresponding histogram. Lastly, what’s about the phospho/total ratio ?
- Lines 327-328: “[…] in absence of taurine, a slight decrease of PGC-1α and MEF2C levels was observed in the extracts of old mice compared to young ones […]”. This point should be tempered. According to figure 3A, PGC-1α remains unmodified while MEF2C seems to decrease whereas not significant.
Author Response

(The authors gave the same response as above.)
